# Effects of Maternal Low-Protein Diet on Microbiota Structure and Function in the Jejunum of Huzhu Bamei Suckling Piglets

**DOI:** 10.3390/ani9100713

**Published:** 2019-09-23

**Authors:** Jipeng Jin, Liping Zhang, Jianlei Jia, Qian Chen, Zan Yuan, Xiaoyan Zhang, Weibo Sun, Cunming Ma, Fafang Xu, Shoujun Zhan, Limin Ma, Guihua Zhou

**Affiliations:** 1College of Animal Science and Technology, Gansu Agricultural University, Lanzhou 730070, China18219613915@163.com (X.Z.); swb887246@126.com (W.S.); 2Key of Laboratory of Plateau Ecology and Agriculture, Qinghai University, Xining 810016, China; 3College of Agriculture and Animal Husbandry, Qinghai University, Xining 810016, China; a18740958061@163.com (Q.C.); YuanZan13897405406@163.com (Z.Y.); 4Qinghai Province Huzhu County Bamei Pig Seed Breeding Farm, Huzhu, Qinghai 810500, Chinazsj632126@163.com (S.Z.); mmm_258369@yeah.net (L.M.); mmm_147258@126.com (G.Z.)

**Keywords:** maternal low-protein diet, 16S rRNA, jejunum microbiota, suckling piglets, bioinformatics

## Abstract

**Simple Summary:**

In mammals, the intestine is the main organ where nutrients are digested and absorbed, thus serving a central role in the growth and health of animals. Low-protein diets are based on the “ideal amino acid” theory, where AA crystals are added to ensure the amino acid balance of the diet. As a result, the amino acid nutritional needs of piglets are reduced along with the total protein level of the diet as well as the metabolic burden of piglets. This typically results in reduced diarrhea and improved gut health. In this study, the microbial composition and function of jejunum chyme samples were analyzed from 30 Huzhu Bamei suckling piglets from sows fed diets with different protein levels via16S rRNA sequencing and bioinformatics. This work provides a theoretical basis for the production of low-protein diets for use in the production of Bamei pigs.

**Abstract:**

The jejunum is the primary organ for digestion and nutrient absorption in mammals. The development of the jejunum in suckling piglets directly affects their growth performance post-weaning. The jejunum microbiome plays an important role in proliferation, metabolism, apoptosis, immune, and homeostasis of the epithelial cells within the organ. The composition and diversity of the gut microbiome is susceptible to the protein composition of the diet. Therefore, the effects of maternal low-protein diets on piglets’ intestinal microbial structure and function have become a hot topic of study. Herein, a maternal low-protein diet was formulated to explore the effects on jejunum microbiome composition and metabolic profiles in Bamei suckling piglets. Using 16S ribosomal RNA (16S rRNA) sequencing in conjunction with bioinformatics analysis, 21 phyla and 297 genera were identified within the gut microflora. The top 10 phyla and 10 genera are within the gut bacteria. Next, KEGG analysis showed that the low-protein diet significantly increased the gut microbial composition, transport and catabolism, immune system, global and overview maps, amino acid metabolism, metabolism of cofactors and vitamins, endocrine system, biosynthesis of other secondary metabolites, signal transduction, environmental adaptation, and cell motility. Taken together, low-protein diets do not appear to affect the reproductive performance of Bamei sows but improved the gut microbiome of the suckling piglets as well as reduced the probability of diarrhea. The data presented here provide new insights on the dietary protein requirements to support the Huzhu Bamei pig industry.

## 1. Introduction

The mammalian intestine is the main organ through which nutrients are digested and absorbed, thus playing a vital role in the growth and health of animals. The maturity of mammalian intestinal development is linked closely to the organism’s absorption function. Nutrients from feed are absorbed along the crypt–villus axis in the gut of newborn animals. In this region, the intestinal microbiota has profound effects on the growth and development of the animal through regulation of intestinal nutrient metabolism, immune system development, and establishment of the intestinal barrier [1,2]. Animal intestines contain a large quantity of extremely complex and diverse microbial communities that play an important role in animal metabolic processing [3,4], energy production [5,6], immune and cognitive development [1], and epithelial homeostasis [7]. The pig breeding industry, as the largest consumer of protein feed, has been required to reduce nitrogen emissions. One way the industry has sought to accomplish this is through low-protein balanced amino acid diets to promote industry health and sustainable development [8]. It is well known that dietary protein quantity and quality can impact the microbial community structure in animals. It has been reported that dietary protein levels have a more significant effect on gut microbial community structure than the source of dietary protein [9]. Chen found that 15% dietary protein restriction can optimize gut microbial community structure through increasing the proportion of beneficial microbes relative to harmful bacterial. As a result, gut tight junction protein functions as well as epithelial cell proliferation were greatly improved [10]. In another study, when sows consumed a diet containing 12.7% protein, the nutritional needs of the suckling piglets were met. Furthermore, limiting amino acids did not reduce milk protein yield or piglet growth rate and concurrently increased the utilization of N, Arg, Leu, Phe+Tyr, and Trp [11].

Dietary proteins have many nutritional and biological functions. In addition to their nutritional role in providing amino acids (AAs) for endogenous protein synthesis, they are also involved in regulating feed intake, lipid and glucose metabolism to sustain growth, development, multiplication of organisms, digestive enzymes, and hormone secretion in the gut [12,13,14]. In pig production, the dietary crude protein (CP) content can be reduced when essential amino acid (EAA) and total nitrogen requirements are met, as dietary protein is merely a source of AAs in pigs [15]. In pig production, piglets enter a critical stage of rapid growth and development in the gastrointestinal tract. As a result, piglets have a high demand for dietary protein and amino acids [16]. Unfortunately, the structure and function of the piglet gastrointestinal tract are not well developed at this period. Changes in the external environment may lead to enhanced stress (from liquid mother’s milk to solid feed), immune dysfunction, and disease susceptibility (pathogenic diarrhea). High-protein diets have often been considered to be important causes of diarrhea and growth inhibition in piglets [8].

Here, the microbial composition and functions of jejunum chyme samples from 30 Huzhu Bamei suckling piglets from sows fed different protein levels were analyzed. A total of 21 phyla and 297 genera were detected within the gut microflora. The top 10 phyla and 10 genera observed based on 16S rRNA sequencing and KEGG analysis indicated that the low-protein diet significantly altered gut microbial composition, transport and catabolism, immune system, global and overview maps, amino acid metabolism, metabolism of cofactors and vitamins, endocrine system, biosynthesis of other secondary metabolites, signal transduction, environmental adaptation, and cell motility, which were significantly increased. Collectively, the data indicated that a low-protein diet will not affect the reproductive performance of Bamei sows but will improve the gut environment of Baimei suckling piglets, reducing the probability of diarrhea. Taken together, the results provide new insights into the dietary protein requirements for the Huzhu Bamei pig industry.

## 2. Materials and Methods

### 2.1. Sample Collection

A total of 30 purebred Huzhu bamei sows from the Qinghai Province Huzhu County Bamei Pig Seed Breeding Farm (Huzhu, China) were selected. The selected sows were approximately the same weight, age, and health status. They were then randomly allocated to one of three treatment groups 14.00% (normal protein, NP), 12.00% (low protein, LP), and 10.00% (very low protein, VLP), with 10 animals per treatment in each pen. Both sows and piglets received identical management and immunization procedures. The feed rations were determined according to the Chinese feeding standard of 90 kg fat per sow. The nutritional composition of each sow’s diet is presented in Table 1. After five days of acclimation, sows were provided with their respective experimental diets and missed one estrous cycle (21 days) during natural estrus and mating on the Qinghai Province Huzhu County Bamei Pig Seed Breeding Farm in Huzhu, Qinghai Province, China. The protocol for animal use for this research was approved by the Gansu Agricultural University’s Academic Committee and the National Natural Science Foundation of China according to guidelines established by the Biological Studies Animal Care and Use Committee of Gansu Province (Approval No. 31660670). From each group, 10 weight-matched piglets were euthanized with 50 mg/kg sodium pentobarbital after 12-h fasting on day 7 of lactation. Both ends of the jejunum were ligated in situ prior to removal of the whole gastrointestinal tract. Contents of the middle jejunum lumen were collected, placed into sterile 1.5 mL polypropylene tubes, and stored in liquid nitrogen for further analysis.

### 2.2. 16S rRNA Sequencing and KEGG Analysis

Sequencing of the 16S rRNA gene was outsourced to BIOMARKER (Beijing, China). Illumina HiSeq 2500 sequencing was used to characterize microbial diversity and community composition. Using the extracted DNA as a template, PCR was performed using barcode primers located on both sides of the V3-V4 hypervariable region of the bacterial 16S rRNA gene. The primer sequences used here were 338F: 5′-ACTCCTACGGGAGGCAGCA-3′ and 806R: 5′-GGACTACHVGGGTWTCTAAT-3′. Amplification was performed for 30 cycles using a DNA thermal Cycler (Bio-Rad, California, USA). The thermocycling profile used here was as follows: 98 °C for 2 min followed by 30 subsequent cycles of 98 °C × 30 s, 50 °C × 30 s, then 72 °C × 1 min, and a final extension at 72 °C for 7 min.

The raw sequencing reads from the original DNA fragments were merged using FLASH v1.2.7 [17] and assigned to each sample according to their unique barcodes. High-quality reads for bioinformatics analysis were performed, and all effective reads from each sample were clustered into operational taxonomic units (OTUs) based on a 97% sequence similarity according to UCLUST [18]. For alpha diversity analysis, the OTUs were rarified to several metrics, including curves of OTU rank, rarefaction, and Shannon. Then, indices of Shannon, Chao1, Simpson, and ACE were calculated. For beta diversity analysis, principal coordinate analysis (PCoA) and unweighted pair group method with arithmetic mean (UPGMA) were performed using QIIME by weighted uniFrac distance [19].

In addition, PICRUSt was performed on the abundance predictions of the KEGG orthologs and pathways of bacterial communities [20]. Functional annotation and classification of all identified microbiomes were determined by using pathway analyses, which was extracted using the search pathway tool in the KEGG Mapper platform (http://www.genome.jp/kegg/mapper.html). The pathway enrichment statistics were calculated using Fisher’s exact test, and the pathways with a corrected *p* value < 0.05 were considered to be significant pathways.

### 2.3. 16S rRNA Gene Quantification by Relative Quantitative Real-Time PCR

Relative quantitative real-time PCR (qPCR) was performed to determine the 16S copy numbers in *Lactobacillus*, *Streptococcus*, *Actinobacillus*, *Clostridium_sensu_stricto_1*, and *Veillonella*. The universal 16S rRNA gene was used as the internal control. Abundances of genera were determined by relative levels of 16S rRNA expression. The primers used for this assay have been previously published [21,22] (Appendix A). The primers were synthesized by the Suzhou Jinweizhi Biotechnology Co., Ltd (Suzhou, China). The SYBR^®^ Premix Ex TaqTM was purchased from Takara Biomedical Technology Co., Ltd (Beijing, China). The qPCR was performed using a LightCycler 480 Real-time System (Fritz Hoffmann-La Roche Co. Limited, Basel, Switzerland). There were four replicates per sample and control. The PCR reactions and conditions are detailed in Appendix A.

### 2.4. Statistical Analysis

The results of the analyses are presented as mean ± SE, and differences based on the level of dietary protein. The statistical analyses were performed using SPSS, version 21. Duncan’s post hoc test and Kruskal-Wallis rank sum test were used to determine any significant differences between groups. Differences were considered to be significant at *p* < 0.05 and *p* < 0.01.

## 3. Results

### 3.1. Animal Monitoring and Litter Size Analysis

The physical condition of Huzhu Bamei sows was assessed daily throughout the experimental period. Litter size, piglet diarrhea rate, and weight were recorded at the first week. The results showed that litter sizes at birth of the VLP, LP, and NP dietary protein groups was 11.80 ± 0.61, 13.40 ± 0.52, and 12.40 ± 0.40, respectively (Table 2). Compared with NP, litter sizes at birth of LP and VLP group increased by 13.6% (*p* < 0.05) and 5.1% (*p* > 0.05), respectively. The litter viability rate at one day of age were 87.47 ± 4.45%, 85.64 ± 4.82%, and 86.09 ± 2.72%, respectively (*p* > 0.05). Because of the higher litter size, the LP group piglets had lower birth weight than the VLP and NP groups (*p* < 0.05). The piglets’ diarrhea rate at the first week was significantly reduced in the experimental groups (VLP and LP) than control group (NP) (*p* < 0.05).

### 3.2. Maternal Low-Protein Diet Induced Changes to the Jejunum Microbiome

The composition and diversity of the bacterial community in the jejunum of suckling pigs was assessed by sequencing of the 16S rRNA. A total of 1,874,550 high-quality sequences with clean tags were acquired from 10 NP samples, 10 LP samples and 10 VLP samples (Appendix A). Based on 97% species similarity, those sequences were assigned to 561 OTUs, of which 439 OTUs were common among all groups (Figure 1). These OTUs mapped to 21 phyla, 41 classes, 89 orders, 147 families, 297 genera, and 324 species (Appendix A). The Good’s coverage ranged from 0.9991 and 0.9998 indicating that more than 99% of the bacterial taxa were captured from the Bamei piglets (Appendix A). The alpha diversity showed that the number of OTUs in the LP group was significantly higher than was observed in the NP and VLP groups (*p* < 0.05) (Appendix A). This implies that a maternal low-protein diet increased richness and diversity in the piglets’ jejunum (Appendix A). Furthermore, significant differences were observed in richness (ACE and Chao1) between LP and NP (*p* < 0.05) (Appendix A). As can be seen in Appendix A, the microbial diversity in the LP group was highly significant (*p* < 0.01). Thus, a low-protein diet produces the highest richness and diversity.

A total of 21 phyla and 297 genera were identified in the jejunum microbiome. The 10 most abundant phyla and 10 genera are presented in Figure 2. The relative abundances of *Firmicutes*, *Proteobacteria*, *Bacteroidetes*, *Fusobacteria*, *Cyanobacteria*, and *Actinobacteria* were more than 0.1%. *Firmicutes* and *Proteobacteria* were the dominant phyla in all groups. In the LP group (Figure 3A), *Fusobacteria* and *Bacteroidetes* were extremely significantly higher than in the VLP group (*p* < 0.01), *Proteobacteria* was extremely significantly higher than in the NP and VLP groups (*p* < 0.01), and *Actinobacteria* was extremely significantly higher than the NP group (*p* < 0.01). However, *Firmicutes* was extremely significantly lower than was observed in the NP and VLP groups (*p* < 0.01). Relative abundances of each of the 47 genera were more than 0.1%. However, many sequences remained unclassified. As is presented in Figure 3B, *Lactobacillus* in the NP group was extremely significantly higher than the LP and VLP groups (*p* < 0.01), *Actinobacillus* of the LP group was extremely significantly lower than the NP and VLP groups (*p* < 0.01), *Veillonella* of the VLP group was significantly lower than the LP group (*p* < 0.05) and extremely significantly lower than the NP group (*p* < 0.01). However, *Clostridium_sensu_stricto_1* in the VLP group was significantly higher than in the NP and LP groups (*p* < 0.05). *Lactobacillus* and *Veillonella* are the main genera of the intestinal tract, and the results obtained by qPCR corroborated these observations (Figure 3C).

Principal component analysis using the weighted unifrac similarity method revealed that that PC1 and PC2 explained 46.76 and 27.06% of the variation between samples, respectively (Figure 4A). Furthermore, it was determined that the plots of different groups were distinctly separated. Consistent with the PCoA, the jackknifed beta diversity and hierarchical clustering analysis via the UPGMA demonstrated that different groups clustered into their own groups, as can be seen in Figure 4B. In addition, LP and VLP clustered into two subgroups based on PCoA (Figure 4A) and UPGMA (Figure 4B), which was attributed to individual variations of gut microbiome composition.

To investigate the functional capacity of the jejunum bacterial community in the NP, LP, and VLP groups, the PICRUSt method was employed to analyze the KEGG pathways of the jejunum microbiota. As can be seen in Appendix A, the second level KEGG pathway analysis showed that transport and catabolism, immune system, global and overview maps, amino acid metabolism, metabolism of cofactors and vitamins, endocrine system, biosynthesis of other secondary metabolites, signal transduction, environmental adaptation, digestive system, cell motility, and neurodegenerative diseases were increased in the LP group. However, nucleotide metabolism, translation, transcription, replication and repair, immune diseases, drug resistance, carbohydrate metabolism, cancers (overview), and cell growth and death were decreased in the LP group. In addition, the second level KEGG pathways analysis showed that immune system, environmental adaptation, global and overview maps, amino acid metabolism, metabolism of cofactors and vitamins, biosynthesis of other secondary metabolites, transport and catabolism, excretory system, signal transduction, endocrine system, nervous system, and cell motility were increased in the VLP group (Appendix A). However, infectious diseases (bacterial), nucleotide metabolism, transcription, drug resistance, translation, cancers (overview), replication and repair, carbohydrate metabolism, cell growth and death, glycan biosynthesis and metabolism, metabolism of other amino acids, endocrine and metabolic diseases, xenobiotics biodegradation and metabolism, cancers (specific types), folding, sorting, and degradation were decreased in the VLP group.

## 4. Discussion

Maternal low-protein diets have been attributed to a decrease in litter size, resulting in poor growth and increased mortality in piglets [23]. However, the current study showed that the litter size from sows provided a low-protein diet was significantly higher than that of sows receiving a normal or very low protein diet. These observations are in contrast to those of Osgerby [24]. The reduced survival rate and birth weight observed in the VLP group may be attributable to the low protein content in the diet, which deviates from the ideal protein (ideal amino acid) diet model [25]. Alternative nutritional strategies to control piglet diarrhea are crucial and also will not cause security concern, such as reducing dietary CP of weanling piglets. Gut microbiota fermentation of undigested protein and AAs is an important factor contributing to piglet diarrhea [26]. Based on the data presented here, a moderate reduction of dietary protein for sows can reduce the rate of diarrhea in piglets. In addition, a high protein diet has a high acid-binding capacity, which can increase the pH value to a near-middle state, providing a good environment for the proliferation of pathogenic bacteria such as Bacteroides and Clostridium, thereby increasing the incidence of diarrhea in piglets [8].

Recent studies have shown that the distribution of microbes in the gastrointestinal tract of pigs varies among the intestinal segments, as well as between the lumen and mucosa [27]. Compared with the number of microorganisms in the colon (1011–1012 CFU/mL), the number of bacteria in the stomach and small intestine is relatively low. The number of bacteria in the stomach and duodenum was 101–103 CFU/mL, and that in the jejunum and ileum was 104–107 CFU/mL [28]. Relative to the large intestine, less information is available on the microbiota in the stomach and small intestine. The jejunum is the main site of digestion and nutrient absorption. In addition, the development of the jejunum in suckling piglets directly affects growth performance post-weaning. Therefore, it is very important to understand the distribution of jejunal microflora and the spatial changes of intestinal microflora. Studies have shown that the largest number of microorganisms in the body reside in the gastrointestinal tract [29]. For example, the total number of microbial cells in the gut of mammals exceeds 10^14^ [30]. The host’s gut-specific microbial community is affected by heredity, diet, and environment. The gut microbiota plays an important role in host physiology and metabolism. It has also been reported that undigested dietary protein may result in the proliferation of protein-fermenting bacteria [31]. Clearly, numerous factors affect the makeup of the host microbial community. The level of CP in the diet, rather than its source, has a more pronounced effect on the composition of the gut microbiota [9]. In the LP group, the modified diet effectively lowered the incidence of diarrhea, maintained gut health, and exerted a distinct influence on both the gut morphology and microbiota [26]. Similar results were obtained in the present study. In addition, a decrease in the incidence of diarrhea was also observed in the VLP diet group.

Early postnatal life is a key time for the development of the immune system and colonization of the host by microbiota. Recent studies have shown that the immune system of early life is regulated by the microbiota, and colonization of the mucosal tissue controls the ability of the host to interact with the flora. In adults, the gut microbiota not only regulates the host's immune responses, but also participates in the host's metabolism and digestion [32,33]. In the latter case, the flora provides the host mucosa with specific enzymes and other proteins that are otherwise lacking [34]. Six principal phyla generally comprise a healthy gut microbiome: *Firmicutes, Proteobacteria, Bacteroidetes, Fusobacteria, Cyanobacteria*, and *Actinobacteria.* The microbiota structure in neonates is low in diversity and very unstable compared to the adult microbiota [35]. The composition of the microbiota after birth is mainly influenced by dietary and environmental factors. Maternal obesity [36], antibiotic use [37,38], and breast-feeding/formula feeding have been associated with modifications of microbiota composition [39]. As with the results of this study, low-protein diets significantly altered piglet gut microbiota.

In the present study, microbial diversity in piglets born to sows provided a LP diet was highly variable compared to both the NP and VLP diets. These observations are consistent with a previous study showing the structure of the intestinal flora that can be optimized by a moderate protein diet [10]. The microbial diversity, richness, and the microbial community structure within the LP group was significantly different compared to both the NP and VLP groups. Furthermore, the bacterial diversity was significantly different between the NP and VLP groups for the gut. However, the difference in microbial richness was not significant between the NP and VLP groups. Taken together, the data show that nutrition is closely related to gut flora [40,41]. The phyla *Firmicutes* and *Proteobacteria* were dominant in the gut of the suckling Bamei piglets of sows provided low-protein diets. This observation is in agreement with the results from the gut of barrows [10]. In this study, it was observed that the LP diet modified the gut microbiota. The results indicated that *Fusobacteria*, *Bacteroidetes*, *Proteobacteria* and *Actinobacteria* increased their proportions with a 2% reduction in protein content, but the lowest levels of *Firmicutes* was observed in the LP group. For gut *Actinobacteria*, many of the strains have been reported to exhibit anti-tumor activity, and the bioactive substances produced also exhibits antimicrobial activity, which are produced at high levels [42]. Conversely, recent studies have shown that *Veillonella* may contribute to early childhood immune system development [43]. Epidemiological studies of infants have demonstrated that presence of *Veillonella* is negatively correlated with autism [44], asthma [45], and bronchiolitis [46]. In this study, the abundance of *Clostridium_sensu_stricto_1* was highest in the VLP group. However, Fan et al. reported that by decreasing CP from 16% to 10%, levels of *Clostridium_sensu_stricto_1* decreased significantly [47]. These results highlight the importance of protein in the development of the intestinal microbiota.

In recent years, intestinal flora has received great attention. Many preclinical studies have shown that gut bacteria can regulate glucose, lipid, and energy homeostasis through a variety of mechanisms, including changes in gut barrier function, metabolism, and inflammation [48,49]. Furthermore, metabolites produced by the flora regulate the health and disease status of the host [50]. Finally, PICRUSt was used to standardize OTU abundance after the KEGG function prediction analysis. When dietary protein was reduced from 14% to 10%, the second level KEGG pathways analysis showed that transport and catabolism, immune system, global and overview maps, amino acid metabolism, metabolism of cofactors and vitamins, endocrine system, biosynthesis of other secondary metabolites, signal transduction, environmental adaptation, and cell motility increased (Appendix A). Diets with a high protein content have a high buffering capacity and would increase small gut pH, and thereby the proliferation of pathogenic bacteria would likely be favored [51]. The high protein content inevitably leads to an increase of β-hemolytic enterotoxigenic strains of *E. coli,* which can result in toxin mediated damage to the intestines and potentially progress to toxemia [52]. As dietary protein levels decrease, intestinal microbial composition shifts to a higher number of beneficial bacteria, preferentially those that ferment carbohydrates [53]. However, the results of this trial show that low-protein diets reduced the carbohydrate metabolism pathway in Bamei suckling piglets. This could be related to gut dysplasia in lactating piglets. It has been reported that, in a low-protein diet condition, there is an initial increase in *Lactobacilli*, followed by beneficial metabolic activities, which stimulates GIT immunity and the formation of short chain fatty acids (SCFAs) [54,55]. It is well known that SCFAs provide energy, reduce inflammation, and regulate metabolism for intestinal epithelial cells by binding to three G protein–coupled receptors (GPCR) of SCFAs—GPR41 (FFA3), GPR43 (FFA2), and GPR109A [56]. By inhibiting histone deacetylase, intestinal cells can proliferate and differentiate [57]. This process results in production of marker hormones of microflora, thus mediating many functions assigned to microflora through classical endocrine signal transduction [58].

## 5. Conclusions

In the present study, 16S rRNA sequencing and the second level KEGG analysis were used to characterize the effects of low-protein diets on the jejunum microbiota and its biological functions in Huzhu Bamei suckling piglets. Sequence analysis indicated that a low-protein diet significantly altered jejunum microbial composition, while the second level KEGG pathways analysis showed that transport and catabolism, immune system, global and overview maps, amino acid metabolism, metabolism of cofactors and vitamins, endocrine system, biosynthesis of other secondary metabolites, signal transduction, environmental adaptation, and cell motility were increased. Taken together, these data suggest that low-protein diets have the potential to increase the abundance and diversities of the jejunum microbiome and also improve the biological functions of the jejunum microbiome. These observations set the basis for a potentially novel mechanism for the homeostasis of the gut microbiota. A 2% reduction in dietary protein does not appear to negatively affect the reproductive performance of Huzhu Bamei sows. However, the reduction did improve the gut environment of piglets, thus reducing the incidence of diarrhea. In the future, one of the principal challenges will be to study the mechanisms of microbial regulation of dietary protein reduction.

## Figures and Tables

**Figure 1 animals-09-00713-f001:**
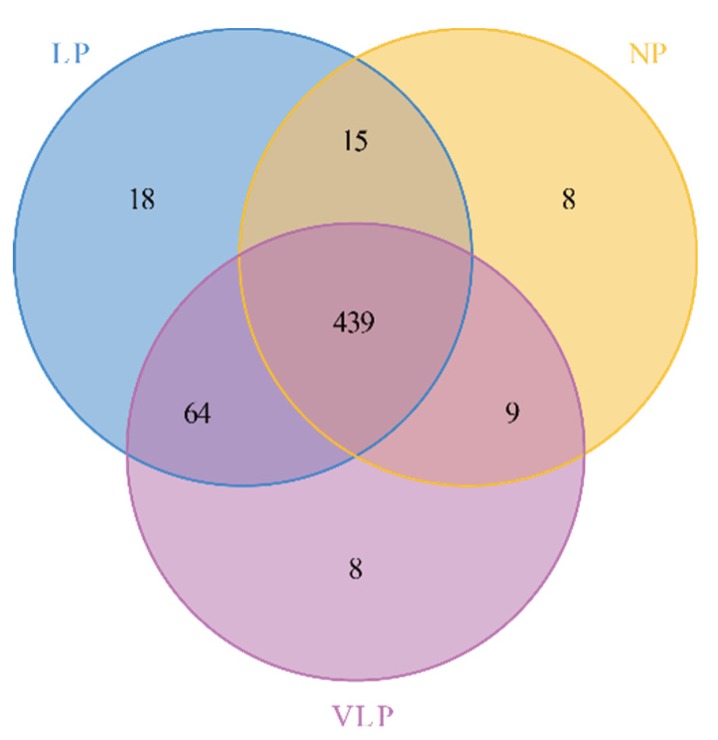
Venn diagram of shared operational taxonomic units (OTUs). The number of overlapping parts is the total number of OTUs among the groups, while the number of non-overlapping parts is the unique number of OTUs among the groups. NP = normal protein; LP = low-protein; VLP = very low-protein.

**Figure 2 animals-09-00713-f002:**
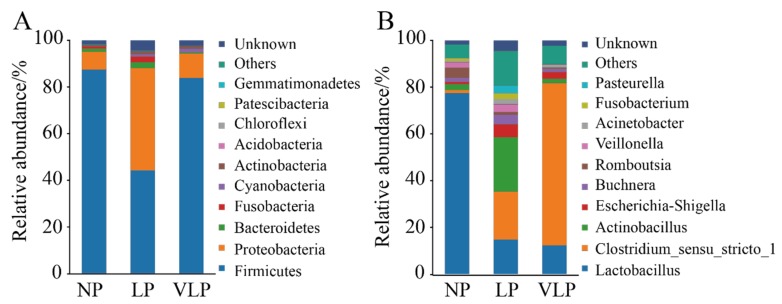
Taxonomic profiles of the gut bacteria in NP, LP, and VLP piglets from 16S rRNA gene sequencing. The relative abundance of the top 10 phylum (**A**) and the top 10 genus (**B**) of gut bacteria present in NP, LP and VLP piglets. NP = normal protein; LP = low-protein; VLP = very low-protein.

**Figure 3 animals-09-00713-f003:**
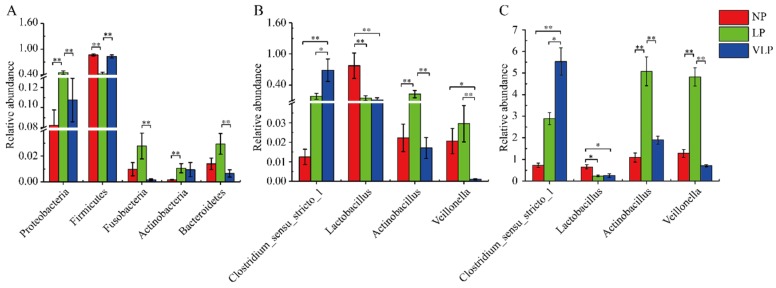
Kruskal-Wallis rank sum test analysis histogram. The bacterial phyla (**A**), genera (**B**), and qPCR (**C**) among NP, LP, and VLP. Bacterial taxa with mean relative abundance greater than 0.1% in least one group are included. Values are expressed as mean ± SE. The symbols represent significant difference among NP, LP, and VLP, ** *p* < 0.01, * *p* < 0.05, NP = normal protein; LP = low-protein; VLP = very low-protein.

**Figure 4 animals-09-00713-f004:**
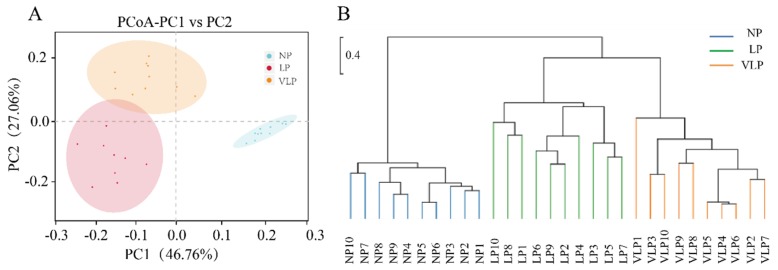
Comparison of gut bacterial community among NP, LP, and VLP. Principal coordinate analysis (PCoA) plot (**A**), UPGMA tree (**B**), all revealing significant differences among NP, LP, and VLP piglets based on the weighted unifrac distances of OTU community. NP = normal protein; LP = low-protein; VLP = very low-protein.

**Table 1 animals-09-00713-t001:** Composition and nutrient levels of diets (DM basis).

Item	Content (%) ^a^		
	VLP	LP	NP
Ingredient composition			
Corn	56.14	50.60	44.90
Soybean meal	0.70	4.50	9.80
Rapeseed Meal	0.00	2.50	2.70
Wheat bran	38.36	37.78	38.14
Lys	0.45	0.34	0.20
Met	0.10	0.07	0.05
Thr	0.21	0.15	0.10
Trp	0.03	0.02	0.01
Val	0.01	0.04	0.10
4% premix ^b^	4.00	4.00	4.00
Nutrient levels			
DE (MJ/kg)	11.71	11.72	11.73
CP	10.00	12.04	14.00
Lys	0.80	0.81	0.80
Met+Cys	0.31	0.33	0.31
Thr	0.36	0.35	0.35
Trp	0.09	0.08	0.08
Val	0.25	0.26	0.23
Total Ca	0.61	0.62	0.62
Total P	0.52	0.51	0.54
Salt	3.20	3.20	3.20

Note: NP = normal protein; LP = low-protein; VLP = very low-protein; DE = digestible energy; CP = crude protein; b The premix during pregnancy provided the following per kilogram of diets: vitamin A: 3.52 kIU; vitamin E: 20 kIU; vitamin D3: 0.76 kIU; vitamin K3: 2.6 mg; vitamin B2: 9.52 mg; vitamin B3: 24 mg; vitamin B5: 45 mg; Cu: 4 mg; Fe: 10 mg; Zn: 40 mg; Mn: 16 mg; Ca: 15%; Total P: 1.8%; NaCl: 8%; Water: 10%.

**Table 2 animals-09-00713-t002:** Effect of protein level on litter sizes in Huzhu Bamei sows.

Item	NP	LP	VLP
Litter size	11.80 ± 0.61 ^b^	13.40 ± 0.52 ^a^	12.40 ± 0.40 ^ab^
Live litter rate (%)	87.47 ± 4.45	85.64 ± 4.82	86.09 ± 2.72
Birth weight (kg)	0.91 ± 0.20 ^a^	0.84 ± 0.20 ^b^	0.90 ± 0.19 ^a^
Diarrhea rate (%)	30.93 ± 13.56 ^a^	20.43 ± 7.27 ^b^	19.90 ± 4.88 ^b^

Noe: In the same row, values with no letter or the same letter superscripts mean no significant difference (*p* > 0.05), while with different small letter superscripts mean significant difference (*p* < 0.05).

## Data Availability

The datasets analyzed during the current study are available from the corresponding author upon request.

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
