# Peer review of "Effects of Maternal Low-Protein Diet on Microbiota Structure and Function in the Jejunum of Huzhu Bamei Suckling Piglets"

_animals, 2019, doi:10.3390/ani9100713_

Round 1

Reviewer 1 Report

Jin et al studied the effects of maternal diets (mainly the protein level) on the microbiota of newborn piglets. This is just rather a descriptive paper. Although the authors attempted to answer an interesting question, there are several shortcomings in the current stage of the manuscript. • Authors observed significant impacts on piglets’ health (especially diarrhea rate) in the groups of LP and VLP. However, the study did not address the impacts on sows’ health. Similarly, the study did focus only on the microbiota of piglets but not sows. • Authors did not justify well why they targeted on jejunum microbiota rather than other parts of the intestine. • The microbiota analysis was done by 16S rRNA sequencing but the authors did not clarify how they generated functional data, perhaps they use functional prediction tool e.g. PICRUSt. But it was not stated in the Methodology section although it was indicated in the result. Authors should cite the original article of PICRUSt tool. • The most important issue for this manuscript is about writing. Several grammatical errors were observed throughout the manuscript and many irrelevant facts were included in the discussion. Some of them are indicated below. o Line 170: “check group” should be “the control group” (the wrong wording) o Line 192: “LP was extremely significant difference NP and LP…” (grammatical error) o Line 305-306: “Epidemiology studies of infants have demonstrated that presence of….” (I don’t feel this is relevant” • Table 2: small letters were not superscripted. • Authors did not perform any analysis between microbiota data and other health outcome data.

Reviewer 2 Report

The authors demonstrate the impact of protein dietary changes in sow feed on the jejunal microbiota of piglets. This was an interesting study, but in this reviewer’s opinion certain comments should be addressed prior to publications

Major comments

Methods and Material and Discussion: How was the jejunal region of the intestine determined. The jejunum is a long section of gut with indistinct borders between the ileum and duodenum. As well, the microbiota in the distal jejunum (ie near the ileum) could be markedly different form the microbiota in the proximal jejunum (near duodenum). This needs to be further clarified in the methods sections and discussed on its importance with the discussion section. Methods and Material and Discussion: The microbiota was only sampled from feces. The samples of microbiota  were not taken from the ‘tightly adherent bacterial communities’  at the mucosal epithelium.  This area has different microbiological communities and have a markedly different impact  on immune function, intestinal physiology, function and so forth within the host. This need to be clarified in the methods and materials section with a fulsome discussion (in the discussion section). In this author’s opinion the presence of diarrhea was quite for high for a commercial operation. The diarrhea may not only be related to dietary changes within treatment groups. Are the sows vaccinated, treated for parasites etc. This should be discussed, as the authors conclude from the study that diarrhea was associated with only diet changes. The data could be affected by (possible) underlying enteric diseases in the sow Table 1: need add the amount of crude fiber in the diets. This affects microbiological fermentation within the large intestine and this affects piglet performance Table 1: need at the amount of crude fat in the diet as this can affect the amount and presence of diarrhea in piglets. Table 2: authors needs to add a daily (or weekly) weight gain of the piglets from sows with different diets. This a very  important consideration for pig performance. This  should be significantly  addressed in the discussion as well. Discussion: this needs a major written revision and more focused. There are areas that were speculative and in this reviewer’s opinion, was not evidently related to the research. As examples (line 261) a discussion on H+E on jejunal mucosa was added, yet no H+E samples of the jejunum were not measured in the current study.  Line 334: there is a discussion on bowel microbiota, yet no samples of bowel  microbiota were measured in this study.

Minor comment:

There are ‘typos’ and some grammatical awkward sentence structure within the paper. Perhaps employ a professional editing company to further refine the manuscript.

Round 2

Reviewer 1 Report

Authors adequately addressed all comments. So, I agree to accept this manuscript for the publication.

Reviewer 2 Report

The authors addressed my major concerns and the writing has markedly improved.